# Surgical and Oncological Outcomes of Salvage Hepatectomy for Locally Recurrent Hepatocellular Carcinoma after Locoregional Therapy: A Single-Institution Experience

**DOI:** 10.3390/cancers15082320

**Published:** 2023-04-16

**Authors:** Takuya Minagawa, Osamu Itano, Minoru Kitago, Yuta Abe, Hiroshi Yagi, Taizo Hibi, Masahiro Shinoda, Hidenori Ojima, Michiie Sakamoto, Yuko Kitagawa

**Affiliations:** 1Department of Hepato-Biliary-Pancreatic and Gastrointestinal Surgery, School of Medicine, International University of Health and Welfare, Chiba 286-0124, Japan; tminagawa@iuhw.ac.jp (T.M.); masa02114@yahoo.co.jp (M.S.); 2Departments of Surgery, Keio University School of Medicine, Tokyo 160-8582, Japan; dragonpegasus427@gmail.com (M.K.); abey3666@gmail.com (Y.A.); hy0624@gmail.com (H.Y.); kitagawa.a3@keio.jp (Y.K.); 3Department of Pediatric Surgery and Transplantation, Kumamoto University Graduate School of Medical Sciences, Kumamoto 860-8556, Japan; taizohibi@gmail.com; 4Departments of Pathology, Keio University School of Medicine, Tokyo 160-8582, Japan; hojima@a3.keio.jp (H.O.); msakamot@z5.keio.jp (M.S.)

**Keywords:** hepatocellular carcinoma, salvage hepatectomy, locoregional therapy, radiofrequency ablation, transarterial chemoembolization, local recurrence

## Abstract

**Simple Summary:**

For 35 patients with recurrent HCC after primary hepatectomy and 67 patients with recurrent HCC after locoregional therapies, surgical and oncological outcomes were examined. Pathologic review revealed 30 patients with locally recurrent HCC after locoregional therapy (LR-HCC). Background liver function was significantly worse in patients with recurrent HCC after locoregional therapy. Serum levels of AFP and AFP-L3 were significantly higher in patients with LR-HCC. Perioperative morbidities were observed in significantly more patients with recurrent HCC after locoregional therapies. Long-term outcomes of recurrent HCC after locoregional therapies were worse than those after hepatectomy, though there was no prognostic difference according to the recurrence patterns after locoregional therapies. Multivariate analyses showed that prognostic factors for resected recurrent HCC were previous locoregional therapy, multiple HCCs, and portal venous invasion, whereas LR-HCC was not a prognostic factor. In conclusion, salvage hepatectomy for LR-HCC showed worse surgical outcomes but a favorable prognosis.

**Abstract:**

Surgical and oncological outcomes of hepatectomy for recurrent hepatocellular carcinoma (HCC) after locoregional therapy, including locally recurrent HCC (LR-HCC), were examined. Among 273 consecutive patients who underwent hepatectomy for HCC, 102 with recurrent HCC were included and retrospectively reviewed. There were 35 patients with recurrent HCC after primary hepatectomy and 67 with recurrent HCC after locoregional therapies. Pathologic review revealed 30 patients with LR-HCC. Background liver function was significantly worse in patients with recurrent HCC after locoregional therapy (*p* = 0.002). AFP (*p* = 0.031) and AFP-L3 (*p* = 0.033) serum levels were significantly higher in patients with LR-HCC. Perioperative morbidities were significantly more frequently observed with recurrent HCC after locoregional therapies (*p* = 0.048). Long-term outcomes of recurrent HCC after locoregional therapies were worse than those after hepatectomy, though there was no prognostic difference according to the recurrence patterns after locoregional therapies. Multivariate analyses showed that prognostic factors for resected recurrent HCC were previous locoregional therapy (hazard ratio [HR] 2.0; *p* = 0.005), multiple HCCs (HR 2.8; *p* < 0.001), and portal venous invasion (HR 2.3; *p* = 0.001). LR-HCC was not a prognostic factor. In conclusion, salvage hepatectomy for LR-HCC showed worse surgical outcomes but a favorable prognosis.

## 1. Introduction

Hepatocellular carcinoma (HCC) is the fifth most fatal disease in the world, and it is especially prevalent in Eastern Asia [1]. The recurrence rate remains high even after curative treatment is performed. The incidence of intrahepatic recurrence within 2 years after primary resection for primary HCC is almost 70% [2]. It is important to develop an optimal strategy for improving the prognosis. The treatment strategy for HCC is proposed depending on the tumor status and the patient’s liver function. Although several guidelines indicate the staging and recommend optimal treatment for primary HCC [3,4,5,6], there has not been any suggested treatment for recurrent HCC after locoregional therapy such as radiofrequency ablation (RFA) or transarterial chemoembolization (TACE). It has been noted in the Japanese guidelines for HCC that a curative treatment strategy that takes hepatic functional reserve into account should be designed for recurrent HCC after RFA [4]. However, the guidelines fall short on the specifics. In clinical practice, because of impaired liver function or declined performance status of the patient, sequential local therapy tends to be selected even after local recurrence.

In addition, no definitive strategy has been clarified in any guidelines according to patterns of recurrence: multicentric recurrence, intrahepatic metastasis, and local recurrence. Locally recurrent HCC after RFA has been thought to be more invasive and needs extensive liver resection [7,8,9,10,11]. Although 5-year recurrence-free survival (RFS) after salvage surgery for recurrent HCC was reported as 0–33%, 5-year overall survival (OS) was revealed to be 43–67% [9,10,11]. On the other hand, the prognosis of the remaining viable HCC after repeated TACE is not fully understood. It is also still unclear whether salvage surgery for locally recurrent HCC after TACE is beneficial.

In this study, we retrospectively evaluated the clinical characteristics of recurrent HCC after locoregional therapy, in particular, locally recurrent HCC after locoregional therapy (LR-HCC). We also studied the perioperative and oncological outcomes of salvage surgery for LR-HCC.

## 2. Materials and Methods

### 2.1. Study Population

A retrospective review of an HCC database was performed. Consecutive patients who had undergone hepatectomy with curative intent between January 2004 and April 2015 were analyzed. This study only included patients who had undergone curative hepatectomy as the first treatment for recurrent HCC. Patients who had undergone re-hepatectomy for recurrent HCC were analyzed, whereas those who had undergone a second or more hepatectomy for recurrent HCC were excluded because they were at risk of double counting as participants in this study. All patients had a confirmed pathologic diagnosis of HCC. The study was approved by the institutional review board of Keio University School of Medicine (unique number: 20120280) and met the standards of the Declaration of Helsinki and the Ethical Guidelines for Clinical Studies of the Ministry of Health, Labour, and Welfare of Japan. This study was registered with the University Hospital Medical Information Network Center (UMIN000014691).

### 2.2. Diagnostic Criteria of Recurrent HCC

Patients after curative treatment for HCC were routinely managed by the sequential follow-up protocol, which consisted of contrast-enhanced computed tomography (CT) or ethoxybenzyl diethylenetriamine pentaacetic acid-enhanced magnetic resonance imaging (EOB-MRI) of the liver within 3 months after the therapy and thereafter every 3 months. Recurrent HCC was diagnosed based on nodules detected by these imaging studies and/or pathologic examinations, such as needle or excisional biopsy, according to the diagnostic algorithm for HCC proposed by the Liver Cancer Study Group of Japan [12].

### 2.3. Treatment Strategy

The treatment strategy for HCC was mainly according to the Japanese guidelines for HCC [2] and was determined by a cluster conference consisting of gastroenterological physicians, radiologists, pathologists, and hepatobiliary surgeons at Keio University Hospital. The hepatectomy was performed by board-certified hepatobiliary surgeons. RFA was performed by well-experienced gastroenterological physicians using a percutaneous, transhepatic approach guided by ultrasonography. TACE was performed by skilled radiologists.

The indications for salvage hepatectomy were mainly classified into the following categories: technical difficulty of repeated locoregional treatment, tumor thrombus, local recurrence after locoregional therapy, and patient preference. The suitable procedure and approach were selected by experienced hepatobiliary surgeons, taking into account the tumor characteristics and the remnant liver function.

Adjuvant systemic chemotherapy and/or local therapy were not routinely administered, even to patients with recurrent HCC. In cases of extrahepatic recurrence after hepatectomy, multidisciplinary treatment, including systemic chemotherapy, radiation therapy, and hepatic arterial infusion chemotherapy, was chosen depending on the patient and tumor condition.

### 2.4. Definition of LR-HCC and OR-HCC

Among intrahepatic recurrence of HCC, LR-HCC was microscopically defined by pathologists as follows: viable tumor cells adjacent to necrotic tissue due to locoregional therapy; morphological similarity to coagulated necrotic tumor cells, which were evaluated by silver stain especially focused on structure and nuclear atypia. In the case of viable tumor cells left in the targeted area of locoregional therapy, the transitional area from the coagulated necrotic tissue was also evaluated. On the other hand, other types of recurrent HCC (OR-HCC) were defined as recurrent HCC that did not meet the definition of LR-HCC above.

### 2.5. Statistical Analysis

Categorical variables were compared using the chi-square test or Fisher’s exact test, as appropriate. Continuous variables were compared using the Mann–Whitney U test. Survival was analyzed using Kaplan–Meier curves and the log-rank test. OS, RFS, and disease-specific survival (DSS) were calculated using the date of the first operation for recurrent HCC. In the total cohort study, the expected deviation in the patients’ backgrounds between the hepatectomy and locoregional groups was calculated, and DSS was evaluated for their prognoses. The optimum cut-off values of each continuous parameter for RFS were determined using the minimum *p*-values calculated using the log-rank test. Hazard ratios were estimated by univariate and multivariate survival analyses using the Cox regression model. Variables with *p* < 0.10 in the univariate log-rank test were further explored in the multivariate setting. Differences were considered statistically significant at *p* < 0.05. All analyses were performed using the SPSS software program, version 28.0 (IBM Corp., Chicago, IL, USA).

## 3. Results

### 3.1. Patient and Tumor Background

We retrospectively reviewed 273 consecutively resected HCC cases at a single center. After excluding 132 cases that underwent hepatectomy only for the primary HCC and not for the recurrent HCC and 39 cases that underwent repeated liver resections for recurrent HCC, 102 resected recurrent HCC cases were extracted.

The clinicopathological characteristics and comparisons according to the previous treatment modalities are listed in Table 1. A total of 35 cases had recurrent HCC after hepatectomy (17 cases after anatomical resection and 18 cases after non-anatomical resection), and 67 cases had recurrent HCC after locoregional therapy (39 cases after RFA and 28 cases after TACE). There were no differences in liver function and tumor markers between the previous treatment modalities. The number of therapies that had been given before was higher in the locoregional treatment group (*p* < 0.001). Surgical outcomes were not significantly different between the two groups except for postoperative complications. The tumors in the locoregional treatment group were significantly larger (*p* = 0.018). In addition, cirrhosis was observed significantly more frequently in the locoregional treatment group (*p* = 0.002). There were no other differences in pathologic features between the two groups. As for surgical margins, all the tumors considered to be positive were microscopically positive (R1). The median follow-up period was 85 months, and the median OS after surgery for recurrence was 83 months. Both OS and RFS of the locoregional therapy group were significantly worse than those of the hepatectomy group (Figure 1A,B). However, the DSS was not significantly different between the two groups (Figure 1C). At the time of initial recurrence, 19 patients had extrahepatic metastases (Table 1), and those who opted for systemic chemotherapy chose sorafenib or the folinate/uracil/tegafur regimen.

### 3.2. Clinicopathologic Characteristics of Patients with LR-HCC

The recurrence pattern was assessed by the pathologic review: 30 cases were diagnosed as LR-HCC and 37 cases as OR-HCC in the locoregional therapy group. Table 2 shows the clinicopathologic features of patients with LR-HCC. Compared with the OR-HCC group, the LR-HCC group had higher serum levels of AFP and AFP-L3 (*p* = 0.031 and *p* = 0.033, respectively). The incidence of postoperative complications tended to be higher in patients with LR-HCC. Pathologic assessment showed that the incidence of positive surgical margins tended to be higher in these patients. There were no obvious differences between the initial recurrence site and the recurrence pattern in the liver after curative surgery. The prognosis was not different between the LR-HCC and OR-HCC groups (Figure 2A,B).

### 3.3. Characteristics and Prognosis of Patients with LR-HCC after RFA and TACE

Table 3 shows the clinicopathologic differences between RFA and TACE stratified by the previous treatment modalities in LR-HCC. LR-HCC after RFA had higher values of des-γ-carboxy prothrombin (DCP) (*p* = 0.041). There were no other differences between the two groups. The prognosis was not different between the two groups (Figure 3A,B).

### 3.4. Prognostic Factors for the RFS of Recurrent HCC

The univariate and multivariate analyses of RFS for recurrent HCC are shown in Table 4. The optimal cut-off values of tumor markers to assign the patients into the two groups based on the greatest difference in the RFS were 20 ng/mL for AFP (*p* = 0.009), 10% for AFP-L3 (*p* = 0.04), and 40 mAU/mL for DCP (*p* = 0.076) when the minimum *p*-value approach was used (Appendix A). Multivariate analysis revealed that locoregional therapy as the previous treatment, multiple tumors, and portal venous invasion were the prognostic factors of RFS in recurrent HCC.

## 4. Discussion

The present study was designed to investigate the clinical benefit of salvage hepatectomy for LR-HCC. In this study, the incidence of postoperative complications was significantly high, which implies that salvage hepatectomy for LR-HCC was technically demanding. On the other hand, the prognosis of LR-HCC after RFA was comparable to those in previous studies [7,9,10,11]. The prognosis of LR-HCC after TACE was shown to be equivalent to that of LR-HCC after RFA. Taken together, these findings suggest that salvage hepatectomy for LR-HCC had favorable OS despite the high incidence of recurrence after curative surgery. In the multivariate analysis, LR-HCC was not a prognostic factor for RFS. Therefore, considering that multidisciplinary sequential therapies are mostly required for LR-HCC because of its highly malignant potential, surgical intervention should be considered as part of treatments if LR-HCC is resectable. It is important to consider hepatectomy and other local treatments as complementary and not exclusive. The dissociation between a low RFS and a rather high OS reflects the slow progression of the disease and the importance of repeating the treatment. Kishi et al. reported that the number rather than the type of treatment for tumor recurrence was associated with prolonged survival [13].

In general, LR-HCC is reported to have a high malignant potential among recurrent HCC [13]. The mechanism of the aggressive behavior remains unclarified. Some studies have concluded that increased intratumoral pressure caused by RFA may favor intravascular tumor spread. Several studies have documented that some recurrent HCC after RFA exhibit aggressive recurrence patterns as reflected in the rate of positive macroscopic tumor thrombus and more extensive tumor distribution. Especially, LR-HCC after RFA tended to be invasive because of lower differentiation grade, capsule invasion, and vascular invasion. However, the present study showed no evident findings rather than high serum levels of tumor markers compared with the other types of recurrent HCC. Immunohistological or genomic assessment of the tumor and tumor microenvironment might reveal reasonable causes of the aggressive behavior of LR-HCC.

Treatments for recurrent HCC are generally selected based on the same criteria as for primary HCC. Therefore, locoregional therapies are easily used again for recurrent HCC after locoregional therapy due to problems such as impaired liver function. Repeating locoregional treatment for intrahepatic recurrence prolongs patient survival and provides a comparable prognosis after RFA to repeat hepatectomy [14,15,16,17]. However, repeated locoregional treatment may lead to poor prognosis when liver resection may be preferable to locoregional treatment from an oncological standpoint in the case of LR-HCC. Appropriate timing of surgical intervention and the establishment of indications for salvage hepatectomy are warranted.

Even in cases where local treatment has been selected due to unresectable factors, surgical treatment may become possible by reviewing the timing and planning tailor-made procedures. In some cases, improvement of liver function through viral therapy or abstinence from alcohol could help preserve postoperative liver function. Recently, minimally invasive surgeries (MIS) have reduced the amount of abdominal wall destruction, thereby reducing leakage of ascites and pleural effusions [18]. In addition to the magnified view of MIS, the development of simulation technology has made it possible to perform accurate resection of the liver based on the understanding of the precise anatomy [19], and partial anatomical resection is now performed to ensure oncological cure, taking into account the remnant liver function. Furthermore, the number of postoperative complications has decreased due to the standardization of surgical techniques, the establishment of a board certification system [20], and the advancement of medical instruments relating to liver dissection and hemostasis methods, which may have fewer adverse effects on postoperative liver regeneration. These factors may have contributed to the selection of liver resection at the time of recurrence, even for patients who would previously have been considered more suitable for local treatment due to unresectable factors in the present study.

Liver transplantation, especially salvage liver transplantation, is the most promising treatment option for recurrent HCC. A meta-analysis reported that the 5-year survival rate after salvage liver transplantation was 53.9%, which was comparable to that after primary liver transplantation (56.5%) [21]. However, the shortage of donor organs, high medical costs, and contraindications for older patients limit the standardization of this strategy. Therefore, salvage hepatectomy might be an alternative treatment to liver transplantation for recurrent HCC, especially for LR-HCC.

To date, no adjuvant therapies have been shown to have benefits, but there are ongoing clinical trials. The IMbrave 050 trial revealed that atezolizumab plus bevacizumab as adjuvant chemotherapy showed prolonged RFS for patients at high risk of HCC recurrence who underwent locoregional or surgical therapy [22]. Considering that the treatment principle of recurrent HCC is the same as that of primary HCC, recurrent HCC with recurrence risk factors might be a good target of adjuvant therapies, including immune checkpoint inhibitors. Moreover, these therapies have the potential to improve the prognosis of patients with highly malignant LR-HCC.

Cytokeratin 19 (CK19) and epithelial cell adhesion molecule (EpCAM) have been known as prognostic biomarkers for HCC [23,24,25]. They might be useful in considering treatment strategies for resectable recurrent HCC. Neoadjuvant or adjuvant therapies might improve CK19- or EpCAM-positive recurrent HCC.

Our study has several limitations. First, it is a retrospective, single-center, small case-series study conducted by expert hepatobiliary surgeons. In addition, it has potential selection bias because the patients in this study had all undergone hepatectomy. We might have chosen patients with better background liver function and fewer multinodular tumors. In particular, the prognosis of the patients with LR-HCC who were treated using non-surgical therapies was unknown. Second, the starting point for prognostic evaluation was the date of resection of recurrent HCC, and previous therapies were not detailed, resulting in potentially varied patient and tumor characteristics. As a result, the prognosis of LR-HCC could not have been fully evaluated. Therefore, the indications and clinical benefit of salvage hepatectomy for LR-HCC were not directly generalized. Multi-institutional prospective cohort studies are warranted to decrease the influence of the potential bias in this study. However, we believe that the results of this study will support the validity of salvage hepatectomy for LR-HCC in selected patients.

## 5. Conclusions

Our study showed that salvage hepatectomy for LR-HCC after locoregional therapies has potentially favorable oncologic outcomes despite being technically demanding. Surgical treatment should be considered for LR-HCC in selected patients.

## Figures and Tables

**Figure 1 cancers-15-02320-f001:**
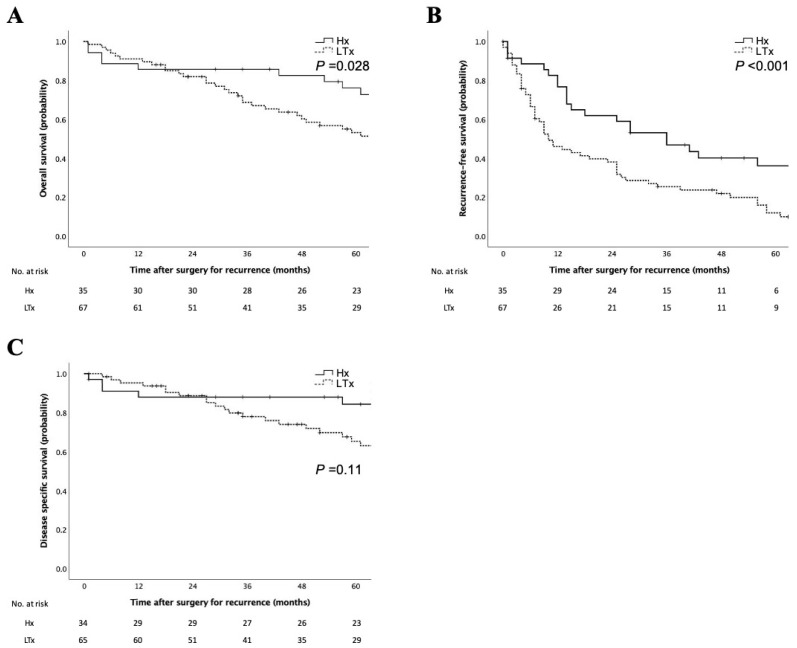
Survival analyses according to the previous treatment modalities. Kaplan–Meier curves for overall survival rates (**A**), recurrence-free survival rates (**B**), and disease-specific survival (**C**) of patients according to the previous treatment modalities. Survival rates in patients with previous locoregional therapy were significantly worse than those in patients with previous hepatectomy in the log–rank test. Hx, recurrence after hepatectomy; LTx, recurrence after locoregional therapy.

**Figure 2 cancers-15-02320-f002:**
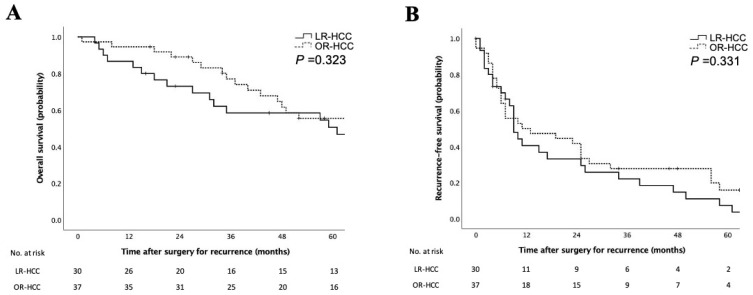
Survival analyses according to recurrence patterns after the locoregional treatment. Kaplan–Meier curves for overall survival rates (**A**) and recurrence-free survival rates (**B**) of patients according to recurrence patterns after the locoregional treatment. Survival rates were comparable between locally recurrent and other types of recurrent hepatocellular carcinoma in the log–rank test. LR-HCC, locally recurrent hepatocellular carcinoma after locoregional therapy; OR-HCC, other types of recurrent hepatocellular carcinoma after locoregional therapy.

**Figure 3 cancers-15-02320-f003:**
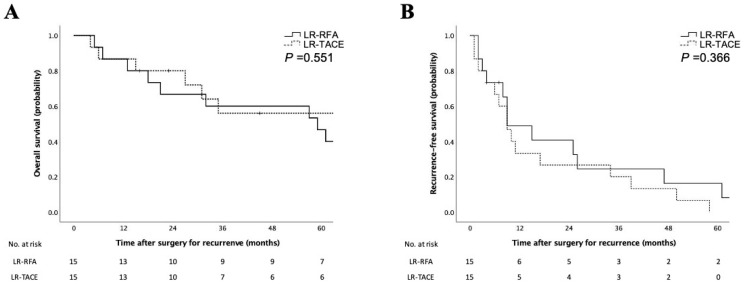
Survival analyses according to the previous locoregional therapy in LR-HCC. Kaplan–Meier curves for overall survival rates (**A**) and recurrence-free survival rates (**B**) of patients according to the previous locoregional therapy in locally recurrent hepatocellular carcinoma. Survival rates were comparable between RFA and TACE before recurrence in the log–rank test. LR-RFA, locally recurrent hepatocellular carcinoma after RFA; LR-TACE, locally recurrent hepatocellular carcinoma after TACE.

**Table 1 cancers-15-02320-t001:** Patient characteristics stratified by the previous treatment modalities.

Variables	Recurrence after Hepatectomy	Recurrence after Locoregional Therapy	*p*-Value
	(*n* = 35)	(*n* = 67)	
Age (year, median)	70 (33–82)	71 (50–86)	0.73
Sex			0.58
Female	9	14	
Male	26	53	
Etiology			0.114
HBV	14	16	
HBV + HCV	0	1	
HCV	12	38	
NBNC	9	9	
Child–Pugh classification			0.296
A	35	63	
B	0	4	
Liver damage			0.206
A	32	53	
B	3	13	
C	0	1	
Platelet count (×10^3^/µL, median)	13.6 (5.1–28.5)	11.8 (4.1–37.4)	0.347
AFP (ng/mL, median)	6 (2–18,000)	10 (0–80,977)	0.286
AFP-L3 (%, median)	7.2 (0–50.3)	9.2 (0–84.9)	0.304
DCP (mAU/mL, median)	21 (9–5220)	25 (7–23,000)	0.956
Number of pretreatments	1 (1–19)	3 (1–10)	<0.001
Surgical approach			0.883
Open	23	45	
Laparoscopic	12	22	
Procedures			0.13
Partial resection	27	44	
Segmentectomy	1	5	
Sectionectomy	6	8	
Hemihepatectomy	1	10	
Operation time (min, median)	291 (118–780)	359 (84–1500)	0.087
Estimated blood loss (g, median)	300 (1–4560)	300 (1–16,156)	0.915
Morbidities (Clavien–Dindo ≥ IIIa)			0.048
No	28	40	
Yes	7	26	
Postoperative hospital stay	13 (6–40)	15 (4–217)	0.313
(day, median)			
Tumor multiplicity			0.236
Solitary	20	30	
Multiple	15	37	
Tumor size (mm, median)	18 (6–42)	20 (7–140)	0.018
Histology			0.429
Well	4	7	
Moderate	23	50	
Poor	7	10	
Other	1	0	
Portal venous invasion			0.532
No	19	32	
Yes	16	35	
Hepatic venous invasion			0.658
No	34	63	
Yes	1	4	
Hepatic arterial invasion			1
No	35	66	
Yes	0	1	
Bile duct invasion			0.658
No	34	63	
Yes	1	4	
Surgical margin			0.678
Negative	26	48	
Positive	7	16	
Background liver condition			0.002
Normal	2	0	
Chronic hepatitis	24	30	
Cirrhosis	7	33	
Initial recurrence site			0.136
Liver	19	41	
Extrahepatic	0	5	
Both	3	11	
Recurrence pattern in liver			0.662
Intrahepatic metastasis/	22	47	
Multicentric occurrence			
Local recurrence	1	5	

Categorical data are expressed as *n* (%). Continuous variables are presented as the median [range]. AFP, alpha-fetoprotein; AFP-L3, lens culinaris agglutinin-reactive AFP; DCP, des-γ-carboxy prothrombin; HBV, hepatitis B virus; HCV, hepatitis C virus; NBNC, non-HBV non-HCV.

**Table 2 cancers-15-02320-t002:** Patient characteristics stratified by the recurrence pattern after locoregional therapy.

Variables	LR-HCC	OR-HCC	*p*-Value
	(*n* = 30)	(*n* = 37)	
Age (year, median)	68 (52–86)	70 (50–80)	0.94
Sex			0.871
Female	6	8	
Male	24	29	
Etiology			0.031
HBV	12	4	
HBV + HCV	0	1	
HCV	14	24	
NBNC	4	8	
Child–Pugh classification			1
A	28	35	
B	2	2	
Liver damage			0.548
A	24	29	
B	6	7	
C	0	1	
Platelet count (×10^3^/µL, median)	11.1 (5.1–19.1)	12.8 (5.1–28.5)	0.57
AFP (ng/mL, median)	35 (4–47,598)	7 (0–314)	0.031
AFP-L3 (%, median)	17.3 (0–84.9)	7.3 (0–50.3)	0.033
DCP (mAU/mL, median)	32 (10–10,520)	19 (7–23,000)	0.897
Number of pretreatments	3 (1–7)	3 (1–10)	0.253
Surgical approach			0.938
Open	20	25	
Laparoscopic	10	12	
Procedures			0.297
Partial resection	16	28	
Segmentectomy	3	2	
Sectionectomy	5	3	
Hemihepatectomy	6	4	
Operation time (min, median)	363 (155–650)	310 (84–1500)	0.123
Estimated blood loss (g, median)	475 (1–2537)	275 (1–16,156)	0.197
Morbidities (Clavien–Dindo ≥IIIa)			0.07
No	14	26	
Yes	15	11	
Postoperative hospital stay	16 (7–43)	12 (4–217)	0.232
(day, median)			
Tumor multiplicity			0.439
Solitary	15	15	
Multiple	15	22	
Tumor size (mm, median)	25 (7–50)	22 (10–80)	0.705
Histology			0.577
Well	3	4	
Moderate	21	29	
Poor	6	4	
Other	0	0	
Portal venous invasion			0.102
No	11	21	
Yes	19	16	
Hepatic venous invasion			0.318
No	27	36	
Yes	3	1	
Hepatic arterial invasion			0.448
No	29	37	
Yes	1	0	
Bile duct invasion			1
No	28	35	
Yes	2	2	
Surgical margin			0.081
Negative	18	30	
Positive	10	6	
Background liver condition			0.268
Normal	0	0	
Chronic hepatitis	16	14	
Cirrhosis	13	20	
Initial recurrence site			0.689
Liver	18	23	
Extrahepatic	3	2	
Both	6	5	
Recurrence pattern in the liver			1
Intrahepatic metastasis/	22	25	
Multicentric occurrence			
Local recurrence	2	3	

Categorical data are expressed as *n* (%). Continuous variables are presented as the median [range]. AFP, alpha-fetoprotein; AFP-L3, lens culinaris agglutinin-reactive AFP; DCP, des-γ-carboxy prothrombin; HBV, hepatitis B virus; HCV, hepatitis C virus; LR-HCC, locally recurrent hepatocellular carcinoma after locoregional therapy; NBNC, non-HBV non-HCV; OR-HCC, other types of recurrent hepatocellular carcinoma after locoregional therapy.

**Table 3 cancers-15-02320-t003:** Patient characteristics stratified by the previous locoregional therapies for LR-HCC.

Variables	Local Recurrence after RFA	Local Recurrence after TACE	*p*-Value
	(*n* = 15)	(*n* = 15)	
Age (year, median)	64 (52–86)	73 (54–79)	0.713
Sex			1
Female	3	3	
Male	12	12	
Etiology			0.513
HBV	6	6	
HCV	8	6	
NBNC	1	3	
Child–Pugh classification			0.483
A	13	15	
B	2	0	
Liver damage			0.651
A	11	13	
B	4	2	
Platelet count (×10^3^/µL, median)	14.2 (5.1–25.8)	11.6 (7.1–29.0)	0.624
AFP (ng/mL, median)	71 (3–47,598)	9 (1–80,977)	0.367
AFP-L3 (%, median)	14.1 (0–84.9)	22.6 (0–58.0)	0.591
DCP (mAU/mL, median)	45 (10–6060)	15 (9–10,520)	0.041
Number of pretreatments	3 (1–5)	3 (1–11)	0.217
Surgical approach			1
Open	10	10	
Laparoscopic	5	5	
Procedures			0.4
Partial resection	8	8	
Segmentectomy	1	2	
Sectionectomy	4	1	
Hemihepatectomy	2	4	
Operation time (time, median)	351 (155–691)	403 (246–801)	0.505
Estimated blood loss (g, median)	400 (1–2537)	510 (1–7100)	0.88
Morbidities (Clavien-Dindo ≥IIIa)			0.858
No	7	7	
Yes	8	7	
Postoperative hospital stay	21 (7–160)	16 (6–101)	0.935
(day, median)			
Tumor multiplicity			0.273
Solitary	9	6	
Multiple	6	9	
Tumor size (mm, median)	20 (7–55)	23 (13–140)	0.806
Histology			
Well	2	1	0.587
Moderate	11	10	
Poor	2	4	
Portal venous invasion			0.256
No	7	4	
Yes	8	11	
Hepatic venous invasion			1
No	14	13	
Yes	1	2	
Hepatic arterial invasion			0.309
No	15	14	
Yes	0	1	
Bile duct invasion			0.483
No	15	13	
Yes	0	2	
Surgical margin			1
Negative	9	9	
Positive	5	5	
Background liver condition			0.34
Normal	0	0	
Chronic hepatitis	7	9	
Cirrhosis	8	5	
Initial recurrence site			0.453
Liver	7	11	
Extrahepatic	1	2	
Both	4	2	
Recurrence pattern in the liver			0.482
Intrahepatic metastasis/	11	11	
Multicentric occurrence			
Local recurrence	0	2	

Categorical data are expressed as *n* (%). Continuous variables are presented as the median [range]. AFP, alpha-fetoprotein; AFP-L3, lens culinaris agglutinin-reactive AFP; DCP, des-γ-carboxy prothrombin; HBV, hepatitis B virus; HCV, hepatitis C virus; NBNC, non-HBV non-HCV.

**Table 4 cancers-15-02320-t004:** Univariate and multivariate analyses of recurrence-free survival.

		Median	Univariate	Multivariate	
Variables	*n*	RFS	*p*-Value	HR (95% CI)	*p*-Value
Etiology			0.208		
HBV	30	26			
HBV + HCV	1	56			
HCV	50	11			
NBNC	21	27			
Child–Pugh classification			0.946		
A	98	17			
B	4	32			
Liver damage			0.404		
A	85	18			
B + C	17	10			
AFP (ng/mL, median)			0.009		0.061
<20	69	25		1 (ref)	
≥20	33	10		1.65 (0.98–2.77)	
AFP-L3 (%, median)			0.04		0.388
<10	81	25		1 (ref)	
≥10	21	10		1.31 (0.71–2.39)	
DCP (mAU/mL, median)			0.076		0.899
<40	67	23		1 (ref)	
≥40	35	13		1.04 (0.60–1.80)	
Number of pretreatments			0.018		0.712
1	47	25		1 (ref)	
≥2	55	11		1.10 (0.67–1.79)	
Pretreatment modality			<0.001		0.005
Hepatectomy	35	36		1 (ref)	
Locoregional therapy	67	10		2.04 (1.24–3.39)	
Tumor multiplicity			<0.001		<0.001
Solitary	50	34		1 (ref)	
Multiple	52	8		2.78 (1.71–4.49)	
Tumor size (mm)			0.029		0.577
≤20	56	25		1 (ref)	
>20	46	14		1.20 (0.66–2.19)	
Portal venous invasion			<0.001		0.001
No	51	25		1 (ref)	
Yes	51	10		2.27 (1.39–3.71)	
Hepatic venous invasion			0.281		
No	97	17			
Yes	5	11			
Hepatic arterial invasion			0.495		
No	101	17			
Yes	1	11			
Bile duct invasion			<0.001		0.19
No	97	19		1 (ref)	
Yes	5	2		1.95 (0.72–5.28)	
Surgical margin			0.049		0.521
Negative	74	25		1 (ref)	
Positive	23	7		1.22 (0.67–2.23)	
Background liver condition			0.411		
Normal/Chronic hepatitis	56	25			
Cirrhosis	40	17			
Recurrence pattern			0.014		0.644
Local recurrence	30	9		1.15 (0.64–2.08)	
Other types of recurrence	72	25		1 (ref)	

Categorical data are expressed as *n*. AFP, alpha-fetoprotein; AFP-L3, lens culinaris agglutinin-reactive AFP; CI, confidence interval; DCP, des-γ-carboxy prothrombin; HBV, hepatitis B virus; HCV, hepatitis C virus; HR, hazard ratio; NBNC, non-HBV non-HCV; ref, reference; RFS, recurrence-free survival.

## Data Availability

Not applicable.

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
