# Peer review of "Surgical and Oncological Outcomes of Salvage Hepatectomy for Locally Recurrent Hepatocellular Carcinoma after Locoregional Therapy: A Single-Institution Experience"

_cancers, 2023, doi:10.3390/cancers15082320_

Round 1

Reviewer 1 Report

Despite its limitations, as expressed by the authors, the work presents interesting results, which partially confirm previous evidence, expressed in a sufficiently clear way.
Did the authors refer to any specific classification for the definition of LR-HCC?If so, please cite it in the text (section 2.4) and in the references.
Considering the conclusions, it would be interesting to include in the discussion a reflection on the reliability of any preoperative criteria (radiological? other?) to define the LR-HCC pattern.

Reviewer 2 Report

Minagawa T et al. reported surgical and oncological outcomes of salvage hepatectomy for locally recurrent hepatocellular carcinoma after locoregional therapy. The results would be valuable in clinical practice. However, there are some concerns that should be addressed.

Major

1.       The author compared the patients who received repeated liver resections with those who underwent resections after locoregional therapy in Figure 1. The background between the two groups was quite different, including the number of pretreatments, tumor size, background liver, and initial recurrence sites. My concern is that Figure 1 could be misleading.

2.       This author should provide explanations for the cut-off levels of tumor markers in Table 4.    

3.       The author only focused on vascular and bile duct invasion. How about the biomarkers already reported, including K19 and EpCAM? The author should reveal the novel findings to make an effective therapeutic strategy for resectable-recurrence HCC.  

4.       Almost half of the patients who received resection after locoregional therapy experienced HCC recurrence within one year. How many patients received systemic therapies after the resection?

5.       The author should discuss adjuvant therapy for the patients who receive resection after locoregional therapy. Now we got the positive results of the IMbrave 050 trial, and the author can presume the high-risk group about postoperative recurrence in patients who received resection after locoregional therapy. 

Reviewer 3 Report

Minagawa et al. demonstrated outcome of salvage hepatectomy for for recurrent hepatocellular carcinoma (HCC) after locoregional therapy using 102 patients' data. This study was interest and contain novelty in this field, and contribute to improvement of our treatment strategy for recurrent HCC. However, there are some point which are needed to revise to improve this study.

Major

#1. in Figure 1A, LR-HCC patients had a poorer prognosis than the other groups, presumably because of the impaired liver function in LR-HCC. have you checked the DSS (disease specific survival)? I think it is misleading to say that LT-HCC has a worse prognosis after salvage liver resection after Hx-HCC without considering DSS that takes liver function into account, and I think you should be careful about stating that.

#2. Of the 102 patients in this study who underwent rehepatectomy, what procedure was used for the 35 patients whose initial treatment was hepatectomy? Was it a anatomical or partial resection? Please add a description of this.

#3. Does surgical margin positive indicate R1 and R2? The frequency of surgical margin positives seems high.

Round 2

Reviewer 2 Report

Thank you for the responses to the comments. There are still some concerns that should be addressed.

1. The author made the new figure (Fig 1C). However, there was no explanation for disease-specific survival. The author should describe the detail. 

2. The author showed the p-value of the cut-off level of tumor markers. However, the author did not describe how to calculate the cut-off level. The author should describe the detail. 

3. The author answered no patient received systemic therapy after liver resection. Did no patient experience extrahepatic metastasis after liver resection? The author should reveal the situation of systemic therapy after liver resection, and this is not an adjuvant therapy. If the patient who received liver resection as LR-HCC experienced extrahepatic metastasis much more than those patients defined as OR-HCC, the finding must be very important. 

Reviewer 3 Report

The authors responded well to reviewers' s comments, therefore, I have no more comments.

Author Response

Thank you for your comment.

Round 3

Reviewer 2 Report

The authors sincerely responded to the reviewer's comments.